# FedRW: Efficient Privacy-Preserving Data Reweighting for Enhancing Federated Learning of Language Models

**Pukang Ye**[*◇]  **Junwei Luo**[♣]

**Jiachen Shen**[◇]  **Saipan Zhou**[◇]  **Shangmin Dou**[△]  **Zhenfu Cao**[◇]  **Hanzhe Yao**[◇]

**Xiaolei Dong**[†◇]  **Yunbo Yang**[†◇▼■]

[◇]East China Normal University  [♣]Wuhan University  [▼]Zhejiang University  [△]ZStack

[■]Hangzhou High-Tech Zone (Binjiang) Institute of Blockchain and Data Security

## Abstract

Data duplication within large-scale corpora often impedes large language models' (LLMs) performance and privacy. In privacy-concerned federated learning scenarios, conventional deduplication methods typically rely on trusted third parties to perform uniform deletion, risking loss of informative samples while introducing privacy vulnerabilities. To address these gaps, we propose Federated ReWeighting (FedRW), the first privacy-preserving framework, to the best of our knowledge, that performs soft deduplication via sample reweighting instead of deletion in federated LLM training, without assuming a trusted third party. At its core, FedRW proposes a secure, frequency-aware reweighting protocol through secure multi-party computation, coupled with a parallel orchestration strategy to ensure efficiency and scalability. During training, FedRW utilizes an adaptive reweighting mechanism with global sample frequencies to adjust individual loss contributions, effectively improving generalization and robustness. Empirical results demonstrate that FedRW outperforms the state-of-the-art method by achieving up to $28.78\times$ speedup in preprocessing and approximately $11.42\%$ improvement in perplexity, while offering enhanced security guarantees. FedRW thus establishes a new paradigm for managing duplication in federated LLM training.

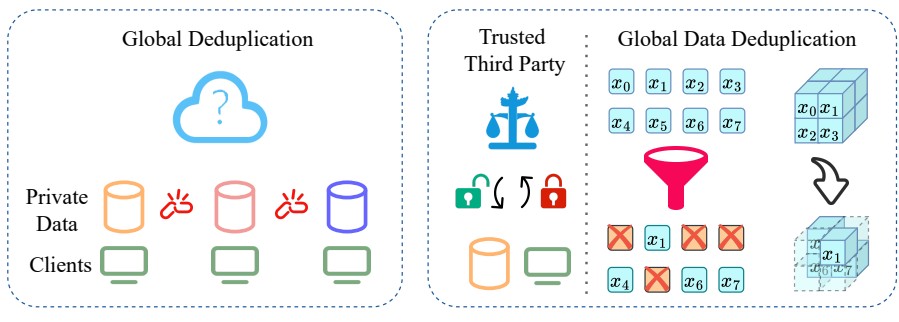

(a) Deduplication challenges in FL  (b) Hard deduplication solution

Figure 1: Deduplication in Federated Learning (FL). (a) Challenges of global deduplication in decentralized settings: privacy constraints prohibit direct data sharing. (b) State-of-the-art solution utilizing hard deduplication over encrypted data, requiring a trusted third party.

---

[*]✉ Main Contact: 51275902028@stu.ecnu.edu.cn

[†]Correspondence authors.

39th Conference on Neural Information Processing Systems (NeurIPS 2025).

# 1 Introduction

Large language models (LLMs) [1–5] have driven remarkable progress across a wide range of applications [6–9]. However, their performance fundamentally depends on data quality, yet real-world corpora often suffer from noise, bias, and especially redundancy. Among these issues, duplicated sequences are particularly widespread in large text datasets [10, 11], weakening generalization and encouraging memorization. This not only hinders downstream performance but also increases vulnerability to privacy attacks such as model inversion, prompt injection, and membership inference [12–15]. As a result, data deduplication has become a standard preprocessing step in training pipelines. Existing techniques fall into two categories: hard deduplication, which removes duplicates via exact or fuzzy matching (e.g., suffix arrays, MinHash) [16, 17]; and soft deduplication, which reweights samples to preserve dataset integrity and avoid brittle thresholding [18–22].

Meanwhile, the growing scarcity of high-quality public data and rising concerns over data privacy [23] have brought federated learning (FL) [24] to the forefront as a compelling alternative for LLM training. By enabling collaborative learning across decentralized clients without local data sharing, FL naturally supports privacy preservation and improved utilization of high-value private data. Yet, FL introduces unique challenges for deduplication, presented in Figure 1(a). Unlike centralized settings, global redundancy across clients cannot be directly resolved due to privacy constraints. A fundamental dilemma emerges: local deduplication fails to detect inter-client duplicates, while global mechanisms cannot bypass privacy silos, leaving redundancy unresolved in federated settings.

Abadi et al. [25]'s EP-MPD represents the most state-of-the-art work for federated hard deduplication, a robust cryptographic framework built on group private set intersection [26], as illustrated in Figure 1(b). Nonetheless, key challenges remain unresolved: (1) strict removal of samples may discard informative or domain-specific content beneficial to model training; (2) multi-round key agreement and encryption introduce significant computational and communication overhead; and (3) reliance on a trusted third party for both encryption and duplicate counting reduces feasibility in stricter privacy settings.

To address the issues mentioned above, we propose Federated ReWeighting (FedRW), to the best of our knowledge, the first framework that enables privacy-preserving soft deduplication in federated LLM training without relying on any trusted third party. Unlike state-of-the-art method that discards duplicated samples, FedRW pioneers a new paradigm of secure, frequency-aware sample reweighting, enabling fine-grained control over sample redundancy while ensuring strict privacy guarantees. At the core of FedRW lies a novel protocol, Privacy-Preserving Multi-Party Reweighting (PPMPR), which securely identifies global duplication patterns across clients through a series of lightweight, third-party-free two-party interactions. To ensure scalability, we further introduce a parallel orchestration strategy that organizes the pairwise interactions into a hierarchical schedule, significantly reducing protocol complexity. Comprehensive experiments demonstrate that FedRW improves both preprocessing efficiency and model generalization, particularly in data-scarce and resource-constrained federated settings. In summary, the key contributions are:

- **FedRW Framework.** Duplicate or overly frequent samples in federated LLM training lead to inefficiency and privacy leakage, especially when deletion-based solutions are impractical. To the best of our knowledge, we propose FedRW, the first framework to achieve privacy-preserving soft deduplication in federated LLM training. Unlike hard deletion methods, FedRW introduces secure, frequency-aware sample reweighting, establishing a new paradigm that bridges privacy protection and data-centric optimization.

- **PPMPR Protocol.** We design PPMPR, a secure protocol for global frequency estimation without relying on a trusted third party. To scale to practical settings, we further introduce a parallel orchestration strategy that reduces the total protocol complexity from $O(n^2)$ to $O(2^{\lceil \log_2 n \rceil})$, achieving 17.61-28.78× acceleration on large datasets and 4.09-28.78× speedup in preprocessing when scaled to 50 parties.

- **Experimental Evaluation.** We conduct extensive empirical studies across diverse datasets and model configurations. By adaptive reweighting, FedRW yields approximately 11.42% perplexity reduction over the baseline, with particularly enhanced robustness under data-scarce and resource-constrained federated settings, where hard deduplication methods often exhibit apparent limitations.

## 2 Related Work

This section reviews data deduplication, categorizing centralized and distributed approaches. We emphasize the limitations in distributed settings, which motivates our proposed FedRW framework.

**Centralized Deduplication.**   Centralized deduplication is crucial for large text corpora, which often contain substantial exact or near-exact samples [10, 11] that degrade model performance and compromise privacy [11, 13–15, 25]. Techniques for exact matching commonly include suffix arrays [16, 27], while fuzzy matching typically employs MinHash for syntactic similarity [11, 16, 17]. Semantic duplication can be identified using pretrained reference models [20, 22, 28].

Instead of removing duplicates, soft deduplication methods reweight training data to mitigate redundancy while preserving the integrity and valuable diversity of datasets. For instance, RedPajama-Data-v2 [29] leverages over 40 quality metrics for systematic filtering and reweighting. DoReMi [20] derives domain-specific weights estimated by a proxy model. Methods like SoftDedup [22] and DSIR [21] quantify sample commonness or importance via n-grams. DrICL [30] uses differentiated learning and cumulative advantages for dynamic reweighting. RHO-1 [31] employs token-level scoring with Selective Language Modeling. However, these centralized strategies are not directly applicable to privacy-concerned FL environments, which effectively leverage high-quality private data.

**Distributed Deduplication.**   Deduplication in FL faces unique challenges due to privacy constraints and data silos. Existing work DupLESS [32] proposes encrypted deduplication using a dual-server architecture, one for encryption key derivation and one for ciphertext deduplication. The state-of-the-art, EP-MPD [25], introduces a group private set intersection framework built on symmetric-key encryption [26] and oblivious pseudorandom functions [33], but still relies on a trusted third party. Critically, these methods focus solely on hard deduplication, neglecting the benefits of reweighting strategies that better preserve data utility and potentially enhance model performance.

These limitations highlight the need for a decentralized soft deduplication solution that ensures privacy without relying on trusted third parties. To this end, we propose an efficient, secure, and third-party-free reweighting framework for federated LLM training, delivering enhanced scalability, performance, and robustness while also ensuring stronger privacy guarantees.

## 3 Preliminaries

**Causal Language Models.**   Causal language models are autoregressive architectures that estimate the joint probability of a token sequence by expressing it into a chain of conditional probabilities:

$$P(x_1, x_2, \ldots, x_n) = \prod_{i=1}^{n} P(x_i \mid x_{<i}), \tag{1}$$

where $P(x_i \mid x_{<i})$ is probability of token $x_i$ given its historical context $x_{<i}$. Model training minimizes the cross-entropy loss to maximize the likelihood of contextually consistent sequences:

$$\mathcal{L} = -\frac{1}{n} \sum_{i=1}^{n} \log P\left(x_i \mid x_{<i}; \theta\right), \tag{2}$$

where $n$ is the sequence length and $\theta$ the model parameters. Perplexity is the standard evaluation metric, calculated as the exponentiated average negative log-likelihood over the sequence:

$$\text{Perplexity} = \exp\left(-\frac{1}{n} \sum_{i=1}^{n} \log P(x_i \mid x_{<i})\right). \tag{3}$$

Lower perplexity signifies reduced prediction uncertainty and better data distribution alignment.

**Security Definition.**   In cryptographic protocol design, the ideal functionality $f$ models the desired behavior of a protocol in an idealized setting. It serves as a trusted third party that collects inputs from all parties, performs the computation securely, and returns the outputs. A protocol is considered secure if its real-world execution is computationally indistinguishable from the ideal execution with $f$. Due to space constraints, formal definitions are deferred to Appendix A.

# 4 Framework

This section details the design and implementation of FedRW. We start by formalizing the PPMPR protocol, followed by a practical construction using cryptographic primitives and a parallel orchestration acceleration strategy for efficiency and scalability. Finally, we describe the integration of the derived weights into the FL training pipeline. An overview of key stages of the FedRW framework is illustrated in Figure 2.

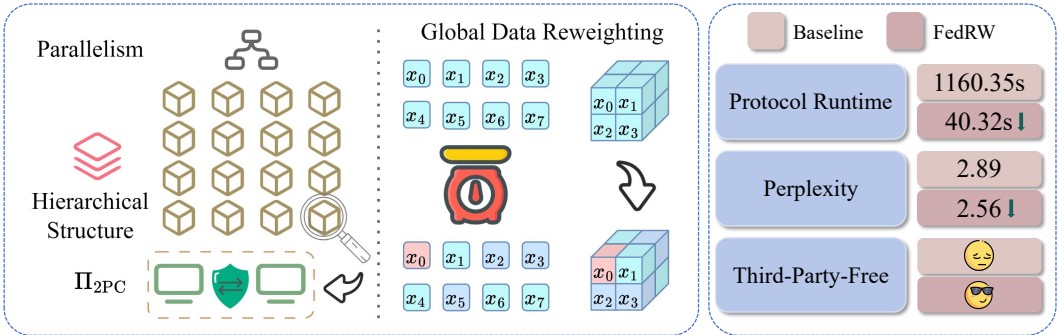

Figure 2: FedRW Framework: Parallel $\Pi_{2PC}$-based Reweighting for Efficient FL. The overview is divided into three parts: **(Left)** The parallel orchestration of the third-party-free $\Pi_{2PC}$ protocol. **(Center)** The frequency-aware reweighting scheme that dynamically assigns weights (reflected by color) to samples while preserving data integrity. **(Right)** A comparison between FedRW and the baseline approach.

## 4.1 Formal definition of PPMPR

Consider a federated setting with $n$ clients $P_1, \ldots, P_n$, where each client $P_i$ holds a local dataset $X_i = [x_1^i, \ldots, x_{m_i}^i]$ consisting of $m_i$ text samples. The objective of our proposed PPMPR protocol is to assign a weight to each sample in a privacy-preserving manner, based on how often it appears across all datasets. This functionality $f_{PPMPR}$ can be formally defined as:

$$f_{PPMPR}(X_1, ..., X_n) \rightarrow (W_1, ..., W_n), \tag{4}$$

where $W_i = [w_1^i, \ldots, w_{m_i}^i]$ refers to the weight vector for the samples in $X_i$. Specifically, each sample $x_j^i$ is associated with an individualized weight $w_j^i$ reflecting its global frequency.

Subsequently, the derived weights are applied to enhance the federated training of LLMs, providing a fine-grained pattern to handle duplicated data. To quantify the relative informativeness of each sample $x$, we employ an intuitive yet effective heuristic: the weight $w(x)$ is inversely proportional to its global frequency:

$$w(x) \propto \frac{1}{freq_{global}(x)}. \tag{5}$$

Here, $freq_{global}(x)$ denotes the occurrence frequency of the sample $x$ within the entire dataset, specifically, the concatenation of all clients' local datasets. This formulation naturally turns the reweighting task into a challenge of securely deriving global frequencies without revealing local data. To solve this problem, we leverage a secure multi-party computation (MPC) approach.

To avoid reliance on a trusted third party, the procedure is decomposed into multiple rounds of secure two-party computation (2PC), a sub-issue of MPC. In a 2PC protocol, two clients, $P_i$ and $P_j$, jointly compute a specific function based on their private inputs, $X_i$ and $X_j$, without directly disclosing the inputs to each other. The functionality $f_{2PC}$ defined for this situation is:

$$f_{2PC}(X_i, X_j) \rightarrow (\vec{C}_i, \vec{C}_j), \tag{6}$$

where $\vec{C}_i$ is vector of length $m_i$, containing the counts of samples in $X_j$ that are identical to each sample $x$ in $X_i$. Since there are no privacy concerns client-side, each unique sample $x$ will be maintained only once in $X_i$, along with its local frequency, $freq_{X_i}(x)$, which can be easily collected and securely shared with other clients that hold the same sample. Through this iterative pairwise 2PC protocol, each client computes and obtains the global frequency for its local samples, allowing them to adjust the sample weights without exposing private data.

## 4.2 Efficient construction of PPMPR

To realize the defined functionalities, we utilize two-party private set intersection (PSI) as the cryptographic foundation of our 2PC protocol. PSI enables two parties to compute the intersecting elements of their datasets without revealing any additional information beyond the agreed-upon rules. The protocol involves only the two participating parties as sender and receiver. In the semi-honest setting, the protocol reveals solely the shared samples and how often they appear in each local dataset, as specified by $f_{2PC}$. The detailed procedure is outlined in Protocol 1.

Table 1: Protocol $\Pi_{2PC}$ in the semi-honest setting model

| **Protocol 1** Two-Party Computation (2PC) | |
| --- | --- |
| **Input:** | Client $P_1$ holds input $X_1 = \{x_1^1, ..., x_{m_1}^1\}$, and client $P_2$ holds input $X_2 = \{x_1^2, ..., x_{m_2}^2\}$. Both input sets are preprocessed local data samples. |
| **Output:** | $P_1$ outputs $\vec{C_1}$, and $P_2$ outputs $\vec{C_2}$, as defined in Eq. (6). |
| **Protocol:** | 1. $P_1$ and $P_2$ initiate a two-party Private Set Intersection (PSI) protocol, where: 
    • $P_1$ acts as sender, and receives nothing. 
    • $P_2$ acts as receiver, and receives the intersection set $\mathcal{I}$ of $P_1$'s data. 
 2. For each sample $x$ in $\mathcal{I}$, $P_2$ extracts the local frequency $freq_{X_2}(x)$, and creates the frequency set $\mathcal{F}_2$. $P_2$ then sends $\mathcal{I}$ and $\mathcal{F}_2$ to $P_1$. 
 3. Upon receiving $\mathcal{I}$ and $\mathcal{F}_2$, $P_1$ extracts the local frequency $freq_{X_1}(x)$, and creates the frequency set $\mathcal{F}_1$. $P_1$ then sends $\mathcal{F}_1$ to $P_2$. 
 4. $P_1$ outputs $\vec{C_1} = [freq_{X_2}(x_1^1), \ldots, freq_{X_2}(x_{m_1}^1)]$, and $P_2$ outputs $\vec{C_2} = [freq_{X_1}(x_1^2), \ldots, freq_{X_1}(x_{m_2}^2)]$. |

The 2PC protocol provides an efficient and secure method for pairwise exchange of sample frequencies between clients. We now extend this building block to construct the full PPMPR protocol.

Table 2: Protocol $\Pi_{PPMPR}$ in the semi-honest setting model

| **Protocol 2** Full Protocol (PPMPR) | |
| --- | --- |
| **Input:** | Each client $P_i$ holds a local dataset $X_i = \{x_1^i, \ldots, x_{m_i}^i\}$, where $i \in \{1, \ldots, n\}$. All datasets are preprocessed. |
| **Output:** | Each $P_i$ outputs a frequency vector $\vec{C_i}$ containing $freq_{global}(x)$ for every $x$ in $X_i$, as defined in Eq. (5). |
| **Protocol:** | 1. Each $P_i$ initialize $\vec{C_i}$ using its local frequencies $freq_{X_i}(x)$ for all $x$ in $X_i$. 
 2. Each $P_i$ performs $\Pi_{2PC}$ once with every other client $P_j$ (all $n-1$ of them). 
    • After each run, $P_i$ outputs $\vec{C_i}$ and updates its global vector: $\vec{C_i} \leftarrow \vec{C_i} + \vec{C_i}$. 
 3. After $n-1$ rounds, $P_i$ outputs the final $\vec{C_i}$. |

As presented in Protocol 2, Each client $P_i$ starts by initializing its frequency vector with local counts, then iteratively executes $\Pi_{2PC}$ with every other clients to progressively build $\vec{C_i}$, the vector of global frequencies. The formal security definitions and proofs are available in Appendix A.

## 4.3 Parallel Acceleration

The formula "N choose 2" represents that the full protocol involves each pair of the $n$ clients performing $\Pi_{2PC}$, which results in $\binom{n}{2}$ executions, leading to an overall time complexity of $O(n^2)$ when run sequentially. This quickly becomes inefficient as a growing number of clients. To address this scalability bottleneck, we introduce a parallel orchestration strategy that reorganizes the execution schedule to minimize overall runtime. We start with a toy example where $n = 8$ in Figure 3, and the detailed procedure is provided in Appendix B.

**Level 3**

| $\Pi_{\text{2PC}}(P_1, P_5)$ | $\Pi_{\text{2PC}}(P_2, P_6)$ | $\Pi_{\text{2PC}}(P_3, P_7)$ | $\Pi_{\text{2PC}}(P_4, P_8)$ |
|---|---|---|---|
| $\Pi_{\text{2PC}}(P_1, P_6)$ | $\Pi_{\text{2PC}}(P_2, P_7)$ | $\Pi_{\text{2PC}}(P_3, P_8)$ | $\Pi_{\text{2PC}}(P_4, P_5)$ |
| $\Pi_{\text{2PC}}(P_1, P_7)$ | $\Pi_{\text{2PC}}(P_2, P_8)$ | $\Pi_{\text{2PC}}(P_3, P_5)$ | $\Pi_{\text{2PC}}(P_4, P_6)$ |
| $\Pi_{\text{2PC}}(P_1, P_8)$ | $\Pi_{\text{2PC}}(P_2, P_5)$ | $\Pi_{\text{2PC}}(P_3, P_6)$ | $\Pi_{\text{2PC}}(P_4, P_7)$ |

**Level 2**

| $\Pi_{\text{2PC}}(P_1, P_3)$ | $\Pi_{\text{2PC}}(P_2, P_4)$ | $\Pi_{\text{2PC}}(P_5, P_7)$ | $\Pi_{\text{2PC}}(P_6, P_8)$ |
|---|---|---|---|
| $\Pi_{\text{2PC}}(P_1, P_4)$ | $\Pi_{\text{2PC}}(P_2, P_3)$ | $\Pi_{\text{2PC}}(P_5, P_8)$ | $\Pi_{\text{2PC}}(P_6, P_7)$ |

**Level 1**

| $\Pi_{\text{2PC}}(P_1, P_2)$ | $\Pi_{\text{2PC}}(P_3, P_4)$ | $\Pi_{\text{2PC}}(P_5, P_6)$ | $\Pi_{\text{2PC}}(P_7, P_8)$ |
|---|---|---|---|

Figure 3: A toy example for the parallel orchestration when $n = 8$.

The key insight is that multiple $\Pi_{\text{2PC}}$ instances can be performed concurrently, provided their participating sets do not overlap. As shown in Figure 3, from the left-hand side of level 1, adjacent pairs of clients perform $\Pi_{\text{2PC}}$ independently. At the next level, these client pairs are grouped into disjoint blocks (e.g., $\{P_1, P_2\}$ with $\{P_3, P_4\}$), and inter-block protocols are executed in parallel. This hierarchical process forms progressively larger blocks, such as $\{P_1, P_2, P_3, P_4\}$ and $\{P_5, P_6, P_7, P_8\}$ at level 3. The structure resembles a binary tree and can be viewed as a recursive two-way merge that manages all $\binom{n}{2}$ sub-protocols efficiently.

To organize this orchestration, the client pairings at each level are structured into pairing matrices, with partial examples highlighted in the dashed-line areas of Figure 3. When $n$ is a power of two, these matrices perfectly arrange all $\Pi_{\text{2PC}}$ executions, maximizing parallelism. Each matrix is constructed by element-wise pairing of two client blocks. For instance, at level 3, matrix $\mathcal{M}_3$ is formed as follows:

$$
\begin{aligned}
\vec{a} &:= (1, 2, 3, 4), \quad \vec{b} := (5, 6, 7, 8) \\
\vec{b'} &\leftarrow \texttt{RotL}(\vec{b}, 0), \quad row_1 \leftarrow \{(\vec{a_i}, \vec{b'_i}) | i = 1, 2, 3, 4\} \\
\vec{b'} &\leftarrow \texttt{RotL}(\vec{b}, 1), \quad row_2 \leftarrow \{(\vec{a_i}, \vec{b'_i}) | i = 1, 2, 3, 4\} \\
\vec{b'} &\leftarrow \texttt{RotL}(\vec{b}, 2), \quad row_3 \leftarrow \{(\vec{a_i}, \vec{b'_i}) | i = 1, 2, 3, 4\} \\
\vec{b'} &\leftarrow \texttt{RotL}(\vec{b}, 3), \quad row_4 \leftarrow \{(\vec{a_i}, \vec{b'_i}) | i = 1, 2, 3, 4\}
\end{aligned}
\tag{7}
$$

Here, $\vec{a}$ and $\vec{b}$ contain the indices of clients from interacting blocks. In each step, $\vec{b'}$ is generated by cyclically left-shifting $\vec{b}$ by $k$ positions using $\texttt{RotL}(\vec{b}, k)$. Client pairs are then formed by matching elements from $\vec{a_i}$ and $\vec{b'_i}$, allowing $\Pi_{\text{2PC}}$ to run concurrently across each row. For $2^{m-1} < n \leq 2^m$, the hierarchical structure remains valid by simply ignoring the unused blocks, thus maintaining optimality and full coverage of client interactions. This parallel approach reduces the total runtime complexity of the full protocol from $O(n^2)$ to $O(2^{\lceil \log_2 n \rceil} - 1)$.

## 4.4 Enhanced Training

To integrate duplication awareness into model optimization, FedRW employs a frequency-based sample reweighting strategy. Given the global frequency vector $\vec{\mathcal{C}}$, where each element represents the occurrence count of a local sample across all clients, the corresponding weight vector $\vec{\mathcal{W}}$ is defined as:

$$
\vec{\mathcal{W}} = \frac{1}{\ln(\vec{\mathcal{C}} + \vec{1}) + \vec{\varepsilon}}
\tag{8}
$$

Here, $\varepsilon$ is a small constant for numerical stability. This formula penalizes frequent samples using a logarithmic function, reducing their impact on optimization without complete exclusion. The logarithm, shifted by 1, ensures that the weights decrease moderately and prevents extreme weights for infrequent samples (e.g., when $\vec{\mathcal{C}}_i = 1$). Compared to linear or hard-threshold formulas, this scheme offers a smoother and adaptive adjustment across varying duplication levels, leveraging the observation that moderate redundancy can promote better model generalization.

These derived weights, $\vec{\mathcal{W}}$, are then applied during training via a sample-wise reweighted loss. Instead of modifying the model architecture, each sample's loss contribution is rescaled by its assigned weight. For a batch of $B$ samples, with $\vec{\mathcal{W}}_i$ as the weight and $\ell_i^{(t)}$ as the token-level average loss of the $i$-th sample, the aggregated batch loss is calculated as:

$$\mathcal{L}_{\text{batch}} = \frac{\sum_{i=1}^{B} \vec{\mathcal{W}}_i \cdot \ell_i^{(t)}}{\sum_{i=1}^{B} \vec{\mathcal{W}}_i} \tag{9}$$

This method diminishes the impact of frequent samples while balancing the influence of less frequent or underrepresented ones. By adapting to statistical redundancy across clients, it preserves informative samples and mitigates overfitting to specific patterns. This provides a lightweight yet effective sample-level reweighting mechanism, particularly advantageous in federated settings with skewed or redundant data. Model updates are then aggregated using the standard FedAvg [24] algorithm.

# 5 Experiments

## 5.1 Experimental Settings

**Environments.** For protocol evaluation, we implement the $\Pi_{2\text{PC}}$ prototype based on [33] and benchmark its runtime under varying configurations. For FL experiments, we use eight public datasets: *Haiku* [34], *Rotten Tomatoes* [35], *Short Jokes* [36], *Poetry* [37], *IMDB* [38], *Sonnets* [39], *Plays*[40], and *Twitter Sentiment Analysis*[41]. To simulate redundancy, duplicates are synthetically added into the training set at different rates and distributed uniformly across 10 clients. The final performance of models is evaluated using perplexity on the test sets. More details can be found in Appendix C.

**Baseline Setting.** We choose EP-MPD [25] as the primary baseline, the most state-of-the-art hard deduplication solution for federated LLM training via a trusted third party. We follow their original experimental settings and directly use their reported runtime and perplexity results for comparison.

## 5.2 Main Results

**Preprocessing.** This part evaluates the efficiency and scalability of our proposed PPMPR protocol against the baseline across three key factors: **dataset size**, **client number**, and **duplication percentage**, with the results shown in tables 3 and 4, and figure 4.

Table 3: Effect of dataset size with $30\%$ duplication percentage on $\Pi_{2\text{PC}}$ running time.

| Method | Protocol Running Time (ms) | | | | | |
|:---:|:---:|:---:|:---:|:---:|:---:|:---:|
| **Dataset Size** | $2^{10}$ | $2^{12}$ | $2^{14}$ | $2^{16}$ | $2^{18}$ | $2^{20}$ |
| Setup | $47.0_{\pm 0.002}$ | $48.6_{\pm 0.003}$ | $54.6_{\pm 0.078}$ | $76.0_{\pm 0.178}$ | $167.8_{\pm 0.478}$ | $715.8_{\pm 1.841}$ |
| Execution | $0.4_{\pm 0.006}$ | $1.0_{\pm 0.006}$ | $5.9_{\pm 0.019}$ | $23.5_{\pm 0.325}$ | $118.7_{\pm 1.738}$ | $713.3_{\pm 7.600}$ |
| $\Pi_{2\text{PC}}$-total | $47.4_{\pm 0.055}$ | $49.7_{\pm 0.118}$ | $60.9_{\pm 0.423}$ | $100.8_{\pm 1.500}$ | $291.8_{\pm 6.250}$ | $1451.8_{\pm 27.141}$ |

The runtime of the basic 2PC protocol increases with dataset size due to the underlying frequency counting mechanism. For small datasets (e.g., $2^{10}$-$2^{14}$), runtime differences are minimal, mainly because the cryptographic setup overhead of two-party PSI is significant compared with the actual execution time, which grows linearly with the dataset size. Noticeably, the execution time scales rapidly beyond a certain dataset size, and begins to dominate the total runtime. For instance, processing $2^{20}$ samples per client takes approximately $1.45$ seconds, as illustrated in table 3.

Table 4 examines how the duplication percentage affects $\Pi_{2\text{PC}}$ runtime when each client holds $2^{19}$ samples. The results show a negligible effect, with the protocol maintaining near-constant performance even at extreme duplication levels. For instance, with 90% duplication, the runtime remains stable at $0.666$ seconds, differing by only $6.9\%$ from the $0.620$-second runtime with 10% duplication. These small variations are attributed to the increased amount of *frequency information exchange* ($\mathcal{F}_1$, $\mathcal{F}_2$ in 3) as the intersection ($\mathcal{I}$ in 3) cardinality grows.

Table 4: Effect of duplication percentage with $2^{19}$ data size in each client on $\Pi_{2PC}$ running time.

| Method | Protocol Running Time (ms) | | | | |
|---|---|---|---|---|---|
| **Duplication Percentage** | 10% | 30% | 50% | 70% | 90% |
| Setup | $342.2_{\pm 1.092}$ | $322.4_{\pm 0.966}$ | $337.4_{\pm 0.988}$ | $339.8_{\pm 1.015}$ | $343.4_{\pm 0.974}$ |
| Execution | $265.9_{\pm 0.259}$ | $293.3_{\pm 3.456}$ | $284.9_{\pm 5.905}$ | $304.2_{\pm 9.283}$ | $310.6_{\pm 11.622}$ |
| $\Pi_{2PC}$-total | $620.1_{\pm 12.213}$ | $626.9_{\pm 13.091}$ | $633.6_{\pm 14.766}$ | $656.5_{\pm 18.170}$ | $665.9_{\pm 18.913}$ |

Figure 4 analyzes the effect of client number on the runtime of the baseline and PPMPR under consistent experimental settings. The baseline proposes two variants that trade off performance and leakage, and we utilize EP-MPD[I], which prioritizes efficiency while introducing more privacy leakage. PPMPR demonstrates superior efficiency and scalability, achieving a $17.61\times$ to $28.78$ speedup over the baseline with $10$ to $50$ clients. This advantage is primarily due to the efficient $\Pi_{2PC}$ protocol as its basic component, with the parallel orchestration strategy, which reduces the overall computational complexity to $O(m-1)$ for $n \in (2^{m-1}, 2^m]$ clients.

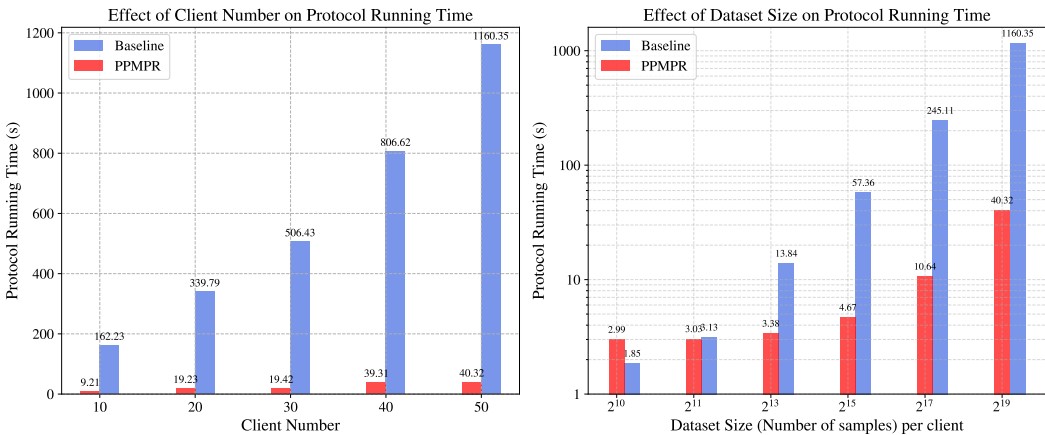

Figure 4: We evaluate the effect of client number and dataset size on protocol running time. For clients ($10$-$50$) with $2^{19}$ data per client and $30\%$ duplication, PPMPR exhibits $17.61$-$28.78\times$ speedup. For $50$ clients, PPMPR outperforms the baseline by $4.09$-$28.78\times$ with increasing dataset size.

Furthermore, we evaluate the impact of dataset size per client with $50$ clients. While PPMPR initially lags EP-MPD[I] on smaller datasets, its parallel strategy quickly becomes dominant as data size scales. With $2^{17}$ samples per client, PPMPR achieves a $23.04\times$ speedup. This dual advantage in scaling across both client counts and data volumes positions PPMPR as a highly efficient and scalable solution for real-world federated environments.

**Model Performance.** This section evaluates model performance across eight text datasets with diverse linguistic structures. To simulate realistic data redundancy in FL, we introduce different levels of artificial duplication ($10\%$, $20\%$, and $30\%$) into the training data. Initially, we assess the robustness of each method under two foundational models utilized in the baseline, GPT-2 Large [42] and DistilGPT2 [43], with perplexity as the evaluation metric.

Table 5: Model perplexity ($\downarrow$) on test set under various duplication settings with GPT-2 Large

| Method | Dataset | | | | | | | | | | | |
|---|---|---|---|---|---|---|---|---|---|---|---|---|
| | Haiku | | | Rotten Tomatoes | | | Short Jokes | | | Sonnets | | |
| **Duplication Percentage** | 30% | 20% | 10% | 30% | 20% | 10% | 30% | 20% | 10% | 30% | 20% | 10% |
| Raw Data | 3.26 | 3.25 | 2.98 | 2.65 | 2.61 | 2.53 | 4.11 | 4.03 | 3.94 | 4.39 | 4.34 | 4.31 |
| Baseline | 2.89 | - | - | 2.21 | - | - | 3.79 | - | - | 4.35 | - | - |
| FedRW (Ours) | **2.56** | **2.67** | **2.69** | **1.61** | **1.63** | **1.64** | **3.15** | **3.17** | **3.17** | **4.07** | **4.26** | **4.26** |

As detailed in Table 5, FedRW consistently outperforms the baseline across all datasets and duplication levels with GPT-2 Large. The improvement is evident on the highly structured *Sonnets* and *Haiku* datasets, where FedRW achieves relative perplexity reductions of up to 6.44% and 11.42% at 30% duplication, respectively. The strict structures of these datasets likely worsen the negative impact of redundancy, highlighting FedRW's ability to preserve content diversity and reduce overfitting through adaptive reweighting.

Similar trends are observed on less structured datasets. For *Short Jokes*, FedRW reduces perplexity from 3.79 to 3.15 under 30% duplication, despite its high lexical diversity. Likewise, on *Rotten Tomatoes*, which is composed of short, opinion-based reviews often prone to duplication, perplexity decreases from 2.21 to 1.61. These results indicate FedRW's effectiveness even when redundancy arises from stylistic repetition.

Furthermore, FedRW exhibits robustness to varying duplication rates. While the baseline's hard filtering yields fixed perplexity (10% to 30% duplication), FedRW maintains stable or slightly improved performance. For instance, perplexity on *Short Jokes* remains constant at 3.17, and on *Haiku*, it decreases from 2.69 to 2.56. These observations align with prior research suggesting that controlled repetition can enhance generalization by reinforcing key training patterns [44]. Instead of discarding duplicates, FedRW adaptively reweights updates to retain informative redundancy, as seen in datasets where increased duplication slightly improves performance. This suggests that effectively managed redundancy can amplify useful linguistic or semantic signals, underscoring FedRW's ability to adapt to varying levels of data noise.

Table 6: Model perplexity (↓) on test set under 30% duplication percentage with DistilGPT2

| Method | Dataset | | | | | | |
|---|---|---|---|---|---|---|---|
| | Haiku | Short Jokes | Rotten Tomatoes | IMDB | Poetry | Sonnets | Plays |
| Raw Data | 3.70 | **2.07** | 1.78 | 7.17 | 2.84 | 5.87 | 15.07 |
| Baseline | 3.67 | **2.07** | 1.77 | 7.25 | 3.01 | 6.08 | 16.09 |
| FedRW (Ours) | **3.65** | 2.08 | **1.75** | **7.00** | **2.66** | **5.75** | **14.50** |

To evaluate FedRW's generalizability in resource-limited scenarios, we evaluate it with DistilGPT2, a smaller version of GPT-2 suitable for FL with limited computational resources. Despite its reduced size, which makes it more vulnerable to the negative effects of data duplication, Table 6 shows that FedRW consistently maintains or slightly improves performance across various datasets.

On datasets like *Haiku* and *Short Jokes*, perplexity remains similar across the three methods. However, more noticeable variances emerge on *Sonnets*, *Poetry*, and *Plays*, where the baseline sometimes underperforms even the undeduplicated data. This could be due to the literary structure and the sparse samples of these datasets. As noted in the baseline, hard deduplication considerably reduces the training samples (e.g., *Poetry*: 526 to 405; *Plays*: 542 to 417), potentially increasing training variance, especially for distilled models. By contrast, FedRW's flexible and adaptive approach aims to retain useful instances when handling excessive redundancy. This reweighting strategy provides a more stable training signal to preserve the integrity of sparse datasets, leading to improved generalization.

Table 7: Model perplexity (↓) on test set under 30% duplication percentage on mainsteam models

| Model | Method | Dataset | | | | | |
|---|---|---|---|---|---|---|---|
| | | Haiku | Jokes | Rotten | Poetry | Sonnets | Plays |
| Qwen3-0.6B | Baseline | 2.47 | 2.61 | 1.71 | 2.54 | 4.07 | 8.21 |
| | FedRW (Ours) | **2.36** | **2.44** | **1.59** | **2.21** | **3.62** | **7.23** |
| Qwen2.5-0.5B-Instruct | Baseline | 2.21 | 2.48 | 1.58 | 2.28 | 4.11 | 11.77 |
| | FedRW (Ours) | **2.12** | **2.36** | **1.55** | **2.03** | **3.84** | **9.92** |
| Llama-3.2-1B-Instruct | Baseline | 2.14 | 2.34 | 1.65 | 2.39 | 4.11 | 18.35 |
| | FedRW (Ours) | **2.09** | **2.21** | **1.54** | **1.99** | **4.00** | **16.03** |

To further validate FedRW's applicability beyond the GPT-2 family, we evaluate the performance on three representative modern models with diverse architectures: Qwen3-0.6B [45], Qwen2.5-0.5B-Instruct [46], and Llama-3.2-1B-Instruct [47]. The results in Table 7 demonstrate that data redundancy remains a substantial challenge even for these contemporary architectures. FedRW robustly maintains its advantage in mitigating the impact of redundancy on model performance, particularly under challenging conditions such as data complexity or sparsity. For instance, FedRW achieves an average relative perplexity reduction of approximately 13.43% on the *Plays* dataset across the three models.

Table 8: Model perplexity ($\downarrow$) on test set under 30% duplication percentage on larger models

| Model | Method | Dataset | | | | | | |
|---|---|---|---|---|---|---|---|---|
| | | Haiku | Jokes | Rotten | Poetry | Sonnets | Plays | Twitter |
| Qwen2.5-3B-Instruct | Baseline | 1.69 | 2.09 | 2.20 | 2.33 | 4.14 | 9.17 | 3.35 |
| | FedRW (Ours) | **1.55** | **1.94** | **2.01** | **1.85** | **3.29** | **7.53** | **2.46** |
| Qwen2.5-7B-Instruct | Baseline | 1.68 | 2.07 | 1.74 | 2.09 | 4.52 | 8.82 | 2.24 |
| | FedRW (Ours) | **1.49** | **1.95** | **1.61** | **1.81** | **3.43** | **6.54** | **1.35** |

With increasing model capacity, memorization of specific patterns due to duplication becomes more pronounced and critical, leading to overfitting, degraded generalization, and increased privacy risks [12]. To assess the issue, we conduct experiments on two large-scale models from the Qwen family: Qwen2.5-3B-Instruct and Qwen2.5-7B-Instruct [46]. While larger models may exhibit lower perplexity on certain datasets, the results in Table 8 show that FedRW sustains its performance advantage over the hard deduplication method. Under the extensive near-duplicate contents in *Twitter*, FedRW achieves a relative reduction of approximately 26.57% in perplexity compared to the baseline.

Table 9: Model Perplexity ($\downarrow$) on test set on the Non-IID settings

| Method | IID | Quantity Skew | Label Skew | Feature Skew |
|---|---|---|---|---|
| Baseline | 1.71 | 2.02 | 2.44 | 3.43 |
| FedRW (Ours) | **1.59** | **1.96** | **1.66** | **2.70** |

To evaluate the efficacy of FedRW under Non-IID data distributions, a major challenge in FL, we conduct experiments on Qwen3-0.6B under three scenarios: *Quantity Skew*, *Label Skew*, and *Feature Skew*. For *Quantity* and *Label Skew*, we categorized the *Rotten Tomatoes* dataset by the binary (0/1) labels across 5 clients, with proportions set to [40%, 20%, 20%, 10%, 10%] and label distributions as [(0.5, 0.5), (0.6, 0.4), (0.4, 0.6), (0.9, 0.1), (0.1, 0.9)], respectively. To simulate *Feature Skew*, we allocate *Poetry*, *Sonnets*, and *Plays* to separate clients, as these datasets differ distinctly in terms of text structure, sentence length, and lexical and syntactic complexity. The results in Table 9 confirm FedRW's robustness to provide a stable training process across heterogeneous data distributions.

## 6 Conclusion

In this work, we introduce FedRW, a novel and principled framework designed to tackle the widespread challenge of data duplication in federated language model training. At its core is PPMPR, a secure and efficient protocol for data reweighting. PPMPR enables soft deduplication methods without compromising data privacy or introducing substantial computational and communication costs. Crucially, our protocol works without a trusted third party, enhancing security and achieving notable improvements in efficiency and scalability.

Our comprehensive experiments across diverse text datasets show that FedRW consistently improves model generalization under redundancy, outperforming the state-of-the-art method across varying duplication levels, dataset settings, and model configurations. Beyond simply discarding duplication, FedRW effectively harnesses redundancy to foster more robust representation learning. These compelling results establish FedRW as a practical, privacy-preserving solution for robust federated training in noisy data scenarios. Moreover, its lightweight modular design allows for seamless integration into broader applications, including multimodal learning pipelines and flexible reweighting strategies, highlighting its potential as a fundamental building block for future federated LLM systems.

## Acknowledgements

This work is supported in part by the National Key Research and Development Program of China (Grant No. 2020YFA0712300), in part by the National Natural Science Foundation of China (Grant No. 62132005, 62172162), in part by Shanghai Trusted Industry Internet Software Collaborative Innovation Center, and in part by Fundamental Research Funds for the Central Universities. This work is also supported by the Postdoctoral Fellowship Program of CPSF under Grant Number GZB20250407.

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

# Appendix

## Contents

## A Security

### A.1 Security Proof

Modern cryptographic protocols are typically analyzed under the simulation-based security paradigm, which formalizes security by comparing a protocol's behavior in the real world to that in an ideal world.

In the ideal world, a trusted third party honestly executes the desired functionality. All parties submit their inputs to the trusted third party, and the trusted third party returns the correct outputs to the designated parties. In contrast, the real world involves actual protocol execution among potentially adversaries without a trusted third party.

A protocol is said to be secure if for every adversary in the real world, there exists a simulator in the real world such that no external environment can tell whether it is interacting with a real world or an ideal functionality. This paradigm ensures that the protocol leaks no more information than what is inherently revealed by the output of the ideal functionality.

### A.2 Universal Composability Model

The Universal Composability (UC) [48] framework provides a rigorous model for analyzing the security of cryptographic protocols under arbitrary adversarial conditions. It ensures that a protocol remains secure even when composed with other protocols, making it robust against complex attack scenarios.

In the ideal world, all parties interact through a TTP that computes the desired functionality $f$, ensuring privacy and correctness. In the real world, parties execute a protocol $\Pi$ without a TTP. A semi-honest adversary $\mathcal{A}$ may observe internal states but not deviate from the protocol.

A protocol $\Pi$ is UC-secure if, for any adversary $\mathcal{A}$ in the real world, there exists a simulator $\mathcal{S}$ that produces a view indistinguishable from $\mathcal{A}$'s view in the ideal world. This ensures that the protocol's behavior in the real world is as secure as the ideal world.

### A.3 Threat Model

In this work, we consider a **semi-honest adversary model** in federated learning (FL), where all participants follow the protocol honestly but may attempt to infer additional information from observations. For the scope of this work, we assume no active collusion among parties. While more active or malicious threats—such as inference, backdoor, or reconstruction attacks—exist, these are considered orthogonal to the primary objective of this study.

A protocol $\Pi$ securely computes a functionality $f : \{0,1\}^* \times \{0,1\}^* \rightarrow \{0,1\}^* \times \{0,1\}^*$, where $f = (f_1, f_2)$. For inputs $(x, y)$, outputs $(f_1(x, y), f_2(x, y))$ are returned to respective parties. Extensions to multi-party settings are implied.

A protocol $\Pi$ is secure against semi-honest adversaries if:

**Definition 1** (Security). *For any semi-honest adversary $\mathcal{A}$, there exist probabilistic polynomial-time (PPT) simulators $Sim_1, Sim_2$ such that:*

$$\{Sim_1(x, f_1(x,y))\}_{x,y} \equiv_c \{View_1^{\Pi,\mathcal{A}}(x,y)\}_{x,y}, \tag{10}$$

$$\{Sim_2(y, f_2(x,y))\}_{x,y} \equiv_c \{View_2^{\Pi,\mathcal{A}}(x,y)\}_{x,y}. \tag{11}$$

Here, $Sim_i(w, f_i(x,y))$ denotes a view based on simulator $i$'s input $w \in (x,y)$ and $\Pi$'s output $f_i(x,y)$. $View_i^{\Pi,\mathcal{A}}(x,y)$ represents $\mathcal{A}$'s observation on party $i$'s view during protocol execution. $\equiv_c$ denotes computational indistinguishability, meaning no PPT algorithm can distinguish the two distributions.

## A.4 Formal Definition of Ideal Functionality

We provide the formal definitions of the ideal functionalities employed in Section 4, as detailed in tables 10 to 12.

Table 10: Ideal functionality $f_{\text{Two-Party PSI}}$

| | |
|---|---|
| **Parameters:** | Client $P_1$ holds input $X_1 = \{x_1^1, ..., x_m^1\}$, and client $P_2$ holds input $X_2 = \{x_1^2, ..., x_n^2\}$. |
| **Functionality:** | • Input $X_1 = \{x_1^1, ..., x_m^1\}$ from $P_1$, and $X_2 = \{x_1^2, ..., x_n^2\}$ from $P_2$. |
| | • Output $X_1 \cap X_2$. |

Table 11: Ideal functionality $f_{\text{2PC}}$

| | |
|---|---|
| **Parameters:** | Client $P_1$ holds input $X_1 = \{x_1^1, ..., x_m^1\}$, and client $P_2$ holds input $X_2 = \{x_1^2, ..., x_n^2\}$. |
| **Functionality:** | • Input $X_1 = \{x_1^1, ..., x_m^1\}$ from $P_1$, and $X_2 = \{x_1^2, ..., x_n^2\}$ from $P_2$. |
| | • Output $\vec{C_1}$ and $\vec{C_2}$. |

Table 12: Ideal functionality $f_{\text{PPMPR}}$

| | |
|---|---|
| **Parameters:** | Each client $P_i$ holds a local dataset $X_i = \{x_1^i, ..., x_{m_i}^i\}$, where $i \in \{1, ..., n\}$. |
| **Functionality:** | • Input $X_i = \{x_1^i, ..., x_{m_i}^1\}$ from $P_i$. |
| | • Output $\vec{\mathcal{C}_i}$. |

## A.5 Security of Protocols

**Theorem 1.** $\Pi_{2PC}$ *securely implements the ideal functionality $f_{2PC}$ in the semi-honest model.*

*Proof.* As described in $f_{\text{2PC}}$, we construct a simulator to simulate the behavior of the corrupted party.

**Case 1: $P_1$ is corrupted.**

The simulator $Sim_1$ receives $P_1$'s input $X_1$ and its output from $f_{\text{2PC}}$, which is $\vec{C_1} = [freq_{X_2}(x_1^1), ..., freq_{X_2}(x_{m_1}^1)]$.

1. During the PSI phase, $P_1$ acts as the sender and receives nothing. As PSI protocol securely implements corresponding ideal functionality, then $P_1$ learns nothing about $X_2$ beyond what is revealed by the intersection $X_1 \cap X_2$. $Sim_1$ can simulate this as an empty view with its own inputs $X_1$.

2. $P_1$ receives the intersection set $\mathcal{I}$ and the frequency set $\mathcal{F}_2$ from $P_2$. $Sim_1$ can construct a simulated intersection $\mathcal{I}'$ and a simulated frequency set $\mathcal{F}_2'$ based on $X_1$ and $\vec{C_1}$. For each $x_k^1 \in X_1$:

- If $freq_{X_2}(x_k^1) > 0$ , then $x_k^1$ is added to $\mathcal{I}'$, and its corresponding frequency in $\mathcal{F}_2'$ is set to $freq_{X_2}(x_k^1)$.
- If $freq_{X_2}(x_k^1) = 0$, then $x_k^1$ is not in $\mathcal{I}'$.

The outputs $(\mathcal{I}', \mathcal{F}_2')$ in the ideal world are indistinguishable from the outputs $(\mathcal{I}, \mathcal{F}_2)$ in the real world, as they perfectly match $P_1$'s output $\vec{C}_1$.

3. $P_1$ sends its frequency set $\mathcal{F}_1$ to $P_2$. $Sim_1$ can generate $\mathcal{F}_1$ using $X_1$ and the intersection set $\mathcal{I}$ (or $\mathcal{I}'$).

The view of $P_1$ consists of its input $X_1$, messages sent ($\mathcal{F}_1$), and messages received ($\mathcal{I}, \mathcal{F}_2$). The simulated view $(X_1, \mathcal{I}', \mathcal{F}_1, \mathcal{F}_2')$ is computationally indistinguishable from the real view $(X_1, \mathcal{I}, \mathcal{F}_1, \mathcal{F}_2)$.

**Case 2: $P_2$ is corrupted.**

The simulator $Sim_2$ receives $P_2$'s input $X_2$ and its output from $f_{2PC}$, which is $\vec{C}_2 = [freq_{X_1}(x_1^2), ..., freq_{X_1}(x_{m_2}^2)]$.

1. During the PSI phase, $P_2$ acts as the receiver and receives the intersection set $\mathcal{I}$. Given the security of the PSI protocol, $P_2$ learns nothing about $X_1$ beyond what is revealed by the intersection $X_1 \cap X_2$. $Sim_2$ can construct a simulated intersection $\mathcal{I}'$ based on $X_2$ and $\vec{C}_2$. For each $x_k^2 \in X_2$:

- If $freq_{X_1}(x_k^2) > 0$, then $x_k^2$ is added in $\mathcal{I}'$.
- If $freq_{X_1}(x_k^2) = 0$, then $x_k^2$ is not in $\mathcal{I}'$.

2. $P_2$ sends $\mathcal{I}$ (or $\mathcal{I}'$) and $\mathcal{F}_2$ to $P_1$. $Sim_2$ can perfectly simulate this using $X_2$ and $\mathcal{I}'$.

3. $P_2$ receives $\mathcal{F}_1$ from $P_1$. $Sim_2$ can construct a simulated $\mathcal{F}_1'$ based on $\vec{C}_2$ and $\mathcal{I}'$. For each $x \in \mathcal{I}'$, the corresponding frequency in $\mathcal{F}_1'$ would be $freq_{X_1}(x)$.

The view of $P_2$ consists of its input $X_2$, messages sent ($\mathcal{I}, \mathcal{F}_2$), and messages received ($\mathcal{F}_1$). The simulated view $(X_2, \mathcal{I}', \mathcal{F}_2, \mathcal{F}_1')$ is computationally indistinguishable from the real view $(X_2, \mathcal{I}, \mathcal{F}_2, \mathcal{F}_1)$.

Since the view of both corrupted parties can be simulated given their input and output from $f_{2PC}$, $\Pi_{2PC}$ securely realizes $f_{2PC}$ in the semi-honest model. $\square$

**Theorem 2.** $\Pi_{PPMPR}$ *securely implements the ideal functionality* $f_{PPMPR}$ *in the semi-honest model.*

*Proof.* We construct a simulator $Sim_{PPMPR}$ for a corrupted $P_k$ that receives $P_k$'s input $X_k$ and its final output the global frequency vector $\vec{C}_k$. from the ideal functionality $f_{PPMPR}$

1. $P_k$ initializes $\vec{C}_k$ using its local frequencies $freq_{X_k}(x)$. This is a local computation, and $Sim_{PPMPR}$ can perform the same step.

2. $P_k$ performs $\Pi_{2PC}$ with every other client $P_j$ (for $j \neq k$). After each execution, $P_k$ receives a vector $\vec{C}_k$ and updates $\vec{C}_k \leftarrow \vec{C}_k + \vec{C}_k$.

- For each interaction between $P_k$ and an honest $P_j$, the security proof for $\Pi_{2PC}$ guarantees that a simulator $Sim_{2PC}$ can generate a view for $P_k$ that is indistinguishable from the real view, using only $X_k$ and the output $\vec{C}_k$.
- Since the final $\vec{C}_k$ is the sum of $P_k$'s local frequencies and pairwise learned frequencies, the overall view of $P_k$ is the collection of views with the $n-1$ executions of $\Pi_{2PC}$. $Sim_{PPMPR}$ can invoke $Sim_{2PC}$ for each interaction between $P_k$ and $P_j$ to generate a view to simulate this combination.
- Since $f_{PPMPR}$ only outputs the final $\vec{C}_k$, $Sim_{PPMPR}$ cannot obtain each partial $\vec{C}_k$. However, it can generate intermediate $\vec{C}_k'$ for each interaction such that their sum (plus the initial vector) equals the known final $\vec{C}_k$. Given that $\Pi_{2PC}$ securely reveals only $freq_{X_j}(x)$ for intersecting samples, the exact distribution does not leak additional information to $P_k$ beyond what $f_{PPMPR}$ allows.

3. After $n-1$ rounds, $P_k$ outputs the final $\vec{C}_k$.

The view of $P_k$ consists of its input $X_k$, its initial local frequencies, and the collection of outputs from all $n-1$ pairwise $\Pi_{2PC}$ executions. Since each $\Pi_{2PC}$ is secure against semi-honest adversaries and its view can be simulated, the collection of these simulated views can be combined by $Sim_{PPMPR}$ securely. Therefore, $Sim_{PPMPR}$ constructs a view for $P_k$ that is computationally indistinguishable from its view in a real execution. Thus, $\Pi_{PPMPR}$ securely realizes $f_{PPMPR}$ in the semi-honest model. □

## B  Parallel Ochestration Algorithm

To support the parallel acceleration strategy introduced in Section 4.3, we formally describe the orchestration logic in Algorithm 1. The algorithm organizes client pairs in a structured matrix manner, ensuring that each client performs 2PC protocols with all others while maximizing concurrency. Specifically, it proceeds in $\lceil \log_2 n \rceil$ hierarchical levels, where clients are iteratively grouped into blocks and scheduled to engage in pairwise protocols via index cyclic rotation. The orchestration guarantees correctness while enabling efficient parallelization.

---

**Algorithm 1** Parallel Orchestration for Efficient Execution of PPMPR

1: **Input:** $n$ clients $P_1, \ldots, P_n$ with local datasets $X_1, \ldots, X_n$
2: **Output:** Global frequency vectors $\vec{\mathcal{C}}_1, \ldots, \vec{\mathcal{C}}_n$ for samples of each client
3: Initialize local frequencies: $\vec{\mathcal{C}}_i \leftarrow freq_{X_i}(\cdot)$ for all $i$
4: Let $m \leftarrow \lceil \log_2 n \rceil$        ▷ Total number of levels
5: **for** $l = 1$ to $m$ **do**
6:  Partition clients into $2^{m-l}$ contiguous blocks of equal size
7:  **for all** pairs of blocks $(A, B)$ **do**
8:   Let $\vec{a} \leftarrow$ indices in $A$, $\vec{b} \leftarrow$ indices in $B$
9:   **for** $k = 0$ to $|\vec{b}| - 1$ **do**
10:    $\vec{b'} \leftarrow \texttt{RotL}(\vec{b}, k)$      ▷ Left-rotate indices in $\vec{b}$
11:    **for** $i = 1$ to $|\vec{a}|$ **do**
12:     **in parallel:** run $\Pi_{2PC}(P_{\vec{a}_i}, P_{\vec{b'}_i})$ to update $\vec{\mathcal{C}}_{\vec{a}_i}, \vec{\mathcal{C}}_{\vec{b'}_i}$
13:    **end for**
14:   **end for**
15:  **end for**
16: **end for**
17: **return** $\{\vec{\mathcal{C}}_1, \ldots, \vec{\mathcal{C}}_n\}$

---

## C  Experimental Details

**Datasets.** In this section, we summarize the detailed information of each dataset used in the experiment. As illustrated in Table 13, these datasets span diverse text domains and reflect a wide range of structural and lexical properties. The table presents the source, sample size, average sequence length, and a brief description for each dataset.

Table 13: Basic information of experimental datasets

| Dataset | # Samples | Avg. Sequence Length | Description |
|---|---|---|---|
| Haiku [34] | 15,281 | 100 | Short-form structured 3-line poems |
| Short Jokes [36] | 231,657 | 100 | Concise User-written short jokes |
| Rotten Tomatoes [35] | 10,662 | 200 | Movie review snippets expressing sentiment |
| IMDB [38] | 49,999 | 500 | Full-length movie reviews with richer narrative structure |
| Sonnets [39] | 460 | 400 | William Shakespeare's 14-line poems |
| Poetry [37] | 573 | 1000 | Modern and classic free-form poems by various authors |
| Plays [40] | 521 | 1000 | Dramatic scripts from William Shakespeare with dialogic structure |
| Twitter [41] | 74,000 | 50 | Tweets labeled with sentiment categories |

For all datasets, we adopt a standard 80/20 train/test split. For movie review datasets, only the review texts are retained, and the sentiment labels are discarded during training. For the *Short Jokes* dataset, we randomly sample 50,000 entries to ensure tractable training time across 10 clients. In cases where

datasets such as *IMDB* already contain a predefined test set, we merge the original training and test partitions, shuffle the combined set, and then re-split it according to the 80/20 ratio.

**Environments.** We conduct all secure protocol procedures, including $\Pi_{\text{2PC}}$ and $\Pi_{\text{PPMPR}}$, on a virtualized server equipped with a 4-core Intel Xeon 2.20GHz CPU and 32GB RAM. For model training, we utilize a machine with a 20-core Intel Xeon Platinum 8457C CPU, 200GB RAM, and an NVIDIA H20 GPU with 96GB memory. All software is executed under the Linux environment. Each experiment in preprocessing is repeated four times, and we report the average performance for consistency.

**Hyperparameters.** We adopt FedAvg [24] as the underlying federated optimization algorithm. For GPT-2 Large and DistilGPT2, we train each client for 1–2 and 1–5 local epochs, respectively, until convergence, with a total of 3–5 communication rounds. For Qwen3-0.6B, Qwen2.5-0.5B-Instruct, and Llama-3.2-1B-Instruct, we train each client for 2 local epochs with a total of 2-5 communication rounds. For Qwen2.5-3B-Instruct and Qwen2.5-7B-Instruct, we train each client for 1–2 local epochs with a total of 1-2 communication rounds to avoid overfitting. The models are optimized using AdamW [49] with learning rates ranging from $1\text{-}5 \times 10^{-5}$. A linear warm-up schedule is applied, reserving 10% of training steps for warm-up. To stabilize training, we apply $\ell_2$-norm gradient clipping with a threshold of 1.0. The maximum sequence length is set between 50 and 1000, depending on the dataset, and batch sizes range from 2 to 8 with gradient accumulation steps adjusted accordingly to maintain effective batch size.

**Baseline.** We follow the baseline implementation proposed in [25], which proposes a hard deduplication approach by pre-filtering duplicated training samples. Specifically, each client performs local deduplication to remove identical samples, which assumes that redundant data is uniformly detrimental, and the resulting datasets are used to train the model without further adjustment.

To ensure fair comparison, we utilize the official open-sourced code[3] and apply the same preprocessing pipeline and training settings as in FedRW. All datasets, tokenization schemes, model architectures, and evaluation metrics remain consistent across the baseline and our proposed method.

## D    Sensitivity Analysis

The discussion on sensitivity analysis focuses on the learning rate to assess FedRW's robustness. The analysis of epochs is omitted as we typically utilize a small number as standard practice to prevent overfitting.

We evaluated the model perplexity on DistilGPT2 and Qwen2.5-0.5B-Instruct under learning rates of 1e-3, 5e-4, 3e-4, 1e-4, 5e-5, and 3e-5. We selected the *Plays* dataset to investigate FedRW's generalizability, as it exhibited a significant performance gap in the main results .

Table 14: Model perplexity ($\downarrow$) on *Plays* test set across various learning rates

| Model | Method | Learning Rate | | | | | |
|---|---|---|---|---|---|---|---|
| | | 1e-3 | 5e-4 | 3e-4 | 1e-4 | 5e-5 | 3e-5 |
| DistilGPT2 | Baseline | 16.12 | 16.38 | 16.13 | 14.21 | 15.07 | 14.18 |
| | FedRW (Ours) | **14.42** | **14.79** | **14.74** | **13.17** | **14.50** | **12.76** |
| Qwen2.5-0.5B-Instruct | Baseline | 12.86 | 11.07 | 10.63 | 11.50 | 11.77 | 10.67 |
| | FedRW (Ours) | **11.48** | **9.35** | **8.15** | **8.14** | **9.92** | **9.81** |

As shown in Table 14, FedRW robustly maintains its superior performance compared to the baseline and exhibits stable training behavior across the entire range of tested learning rates. This confirms that FedRW's advantage is not overly sensitive to the learning rate selection.

---

[3]`https://github.com/vdasu/deduplication`

# E   Future Work

**Advanced Paradigms.**   FedRW's lightweight, modular design enables seamless integration into broader applications, including multimodal learning pipelines and flexible reweighting strategies. Integration with personalized FL (e.g., diverse model architectures or personalization strategies) and dynamic client adaptation (where clients join, leave, or exhibit varying computational capabilities) are also valuable aspects for future research.

**Optimizations.**   Addressing semantic redundancy is a significant issue in large-scale real-world corpora for LLMs. It is prospective to leverage the representation learning capability of transformer-based architectures to extract semantic duplication.

**Adversarial Security.**   FedRW primarily operates under a semi-honest threat model, which is standard and foundational for practical privacy-preserving protocols. Extending FedRW to resist malicious adversaries would be an interesting research direction. This could involve integrating mechanisms like Differential Privacy on sample frequencies or utilizing Zero-Knowledge Proofs to verify client consistency during pre-processing and training. These potential schemes trade off between model accuracy, data privacy, and computational overhead.

