# OpenReview forum: "FedRW: Efficient Privacy-Preserving Data Reweighting for Enhancing Federated Learning of Language Models"
_NeurIPS.cc/2025/Conference — NeurIPS 2025 poster_

### Official Review · Reviewer_y8JQ · 2025-06-06

**Clarity:** 2
**Significance:** 2
**Originality:** 2
**Rating:** 4
**Confidence:** 4

**Summary:**

This paper introduces Federated ReWeighting (FedRW), a privacy-preserving framework designed to address the challenges of data duplication in the federated training of large language models (LLMs). Traditional deduplication methods in federated learning often rely on trusted third parties and involve deleting redundant data, which can lead to the loss of valuable information and introduce privacy risks. FedRW proposes a "soft deduplication" approach that reweights data samples based on their global frequency, rather than deleting them, and does so without requiring a trusted third party.

**Questions:**

See above.

**Ethical Concerns:**

["NO or VERY MINOR ethics concerns only"]

**Final Justification:**

The rebuttal has addressed my concerns. I updated the score accordingly.

**Limitations:**

There is no discussion on limitations in the paper.

**Paper Formatting Concerns:**

None.

**Quality:**

2

**Strengths And Weaknesses:**

Strengths
- The paper focuses on the data duplication issue in federated learning, an important topic in the field.
- The paper clearly illustrates the framework in the method part.

Weaknesses
- The paper does not clearly show the difference between the proposed method and the baseline, EP-MPD.
- The paper does not explain the FL experiment settings. What FL settings are used currently? Is the proposed method still effective under different FL settings, such as IID, non-IID, or personalized FL?
- In the experiments, only GPT-2 Large and DistilGPT-2 are used for the method. It would be better to include some more recent LLMs in the experiments.

---

> ### Author Rebuttal · Authors · 2025-07-30
>
> We appreciate the reviewer for the valuable and constructive comments. We are encouraged by the positive feedback on the **significance of topic** and the clarity of the paper. We hope the following discussions can address the reviewer's remaining questions.
>
> **1. Clarity of Difference From Baseline and FL Experimental Settings**
>
> We appreciate this feedback on the clarity of the difference between FedRW and the baseline (EP-MPD), and the FL experimental settings. We want to clarify that we have detailed the key information in our manuscript:
>
> + **Section 1 (Introduction)**: We introduced EP-MPD, the SOTA work for federated hard deduplication, and illustrated three key challenges unresolved, followed by the introduction of our proposed method, FedRW. We also presented an overview of EP-MPD and its potential limitations in Figure 1.
> + **Section 2 (Related Work)**: In the subsection on _**Distributed Deduplication**_ in existing work, we discussed the mechanism and limitations of EP-MPD.
> + **Section 4 (Framework)**: This section offered a detailed explanation of FedRW. In Figure 2, we presented the architecture framework of FedRW and provided an intuitive comparison on protocol runtime, perplexity, and third-party reliance. We consider these presentations to be sufficient in illustrating the essential differences between the two methods.
> + **Section 5 (Experiments)**: The experimental results on preprocessing and model performance clearly showed the difference between FedRW and EP-MPD.
> + **Section 5.1 (Experimental Settings)**: This section declared the datasets, client number, data preprocessing and distribution strategies, evaluation metrics, and overall experimental setup. For completeness, Appendix C includes a detailed description of the datasets, the partitioning standard, the experimental environment, and the hyperparameter configurations.
>
> We sincerely hope that the above clarifications effectively resolve your concerns.
>
>
> **2. Effectiveness Across FL Settings**
>
> We appreciate your concerns for more detailed effectiveness under various FL settings. Regarding the three specific FL settings:
>
> + Current FL settings: In this study, we followed the baseline setup for **unsupervised fine-tuning**. As described in Lines 203-204, "To simulate redundancy, duplicates are synthetically added into the training set at different rates and distributed uniformly across 10 clients." The data follows an **IID** distribution.
>
> + Non-IID: **We appreciate the suggestion to explore the Non-IID setting, and we have conducted experiments on Qwen3-0.6B to evaluate FedRW's effectiveness.**
>   1. Setup 1: We categorized the *Rotten Tomatoes* dataset by *label* and set up 5 clients. To simulate **Quantity Skew**, we distribute the data according to the proportion of the total train set as **[40%, 20%, 20%, 10%, 10%]**. For **Label Skew**, we set the distributions for the binary (0/1) labels as: **[(0.5, 0.5), (0.6, 0.4), (0.4, 0.6), (0.9, 0.1), (0.1, 0.9)]**.
>
>   2. Result 1 (under 25% duplication)
>
>         | Method         | IID | Quantity Skew | Label Skew |
>         |----------------|-------|-------|-------|
>         | Baseline       |1.71| 2.02 | 2.44 | 1.73 |
>         | FedRW (Ours)   |**1.59**| **1.96** | **1.66** |
>
>   3. Setup 2: To simulate **Feature Skew**, we allocated datasets *Poetry*, *Sonnets*, and *Plays* to separate clients. **These datasets differ distinctly in terms of text structure, sentence length, and lexical and syntactic complexity.**
>
>   4. Result 2
>
>         | Method         | Feature Skew |
>         |----------------|-------|
>         | Baseline       | 3.43 |
>         | FedRW (Ours)   | **2.70** |
>
>     The experimental results indicate that FedRW consistently maintains its performance advantages in the **Non-IID** settings.
>
>
> + Personalized FL:
> We appreciate the suggestion to explore personalized FL. While the current FedRW framework primarily focuses on enhancing a global LLM, we acknowledge that language models benefit significantly from personalization to adapt to individual user preferences. A deeper exploration may involve diverse model architectures or personalization strategies, beyond the scope of this foundational privacy-preserving soft deduplication framework. We consider adapting FedRW to integrate with this pattern a valuable and exciting avenue for future research.
>
>
> **3. Evaluations on Mainstream Models**
>
> **We appreciate your valuable suggestion, and have conducted additional perplexity evaluations on Qwen3-0.6B, Qwen2.5-0.5B-Instruct, and Llama-3.2-1B-Instruct under 30% duplication.**
>
> + Qwen3-0.6B
>
>     | Method         | Haiku  | Short Jokes | Rotten Tomatoes | Poetry | Sonnets | Plays |
>     |----------------|--------|-------------|-----------------|--------|---------|-------|
>     | Baseline       |  2.47  |     2.61    |      1.71       |  2.54  |   4.07  |  8.21 |
>     | FedRW (Ours)   |  **2.36**  |     **2.44**    |      **1.59**       |  **2.21**  |   **3.62**  |  **7.23** |
>
> + Qwen2.5-0.5B-Instruct
>
>     | Method         | Haiku | Short Jokes | Rotten Tomatoes | Poetry | Sonnets | Plays |
>     |----------------|-------|-------------|-----------------|--------|---------|-------|
>     | Baseline       |   2.21    |     2.48        |       1.58          |    2.28    |     4.11    |   11.77   |
>     | FedRW (Ours)   |    **2.12**   |      **2.36**       |         **1.55**        |    **2.03**   |   **3.84**    |   **9.92**   |
>
>
> + Llama-3.2-1B-Instruct
>
>     | Method         | Haiku | Short Jokes | Rotten Tomatoes |  Poetry | Sonnets | Plays |
>     |----------------|-------|-------------|-----------------|--------|---------|-------|
>     | Baseline       |   2.14    |     2.34        |     1.65            |         2.39     |    4.11     |  18.35     |
>     | FedRW (Ours)   |    **2.09**   |      **2.21**       |        **1.54**         |          **1.99**    |     **4.00**    |   **16.03**    |
>
> As shown in the tables, FedRW continues to show its advantage in addressing the impact of duplication on model performance, particularly under challenging conditions such as data sparsity or complexity.
>
> **4. Discussion on Limitations**
>
> We appreciate your concern regarding the lack of a detailed discussion on limitations. While we have discussed FedRW’s boundaries and potential extensions in Section 6 (Lines 284–286), as well as certain limitations within the Experiments section, we commit to providing a clearer and more comprehensive discussion of these aspects in the revised manuscript.
>
> **We sincerely hope our replies have clearly addressed your concerns. We are deeply grateful for the valuable feedback and the broader perspectives you offered, which have helped us better position the significance of this work within the broader research landscape.**

---

> > ### Comment · Reviewer_y8JQ · 2025-08-06
> >
> > Thank you for the detailed rebuttal and it has addressed my concerns. I will update the score.

---

### Official Review · Reviewer_xmuh · 2025-07-02

**Clarity:** 3
**Significance:** 3
**Originality:** 3
**Rating:** 5
**Confidence:** 3

**Summary:**

FedRW has proposed an innovative privacy preserving federated learning framework that replaces trusted third parties with a frequency aware sample reweighting mechanism, securely calculates global sample frequencies using the PPMPR protocol, and combines a layered parallel strategy to reduce preprocessing complexity, achieving a preprocessing acceleration of up to 28.78 times while improving model generalization ability.

**Questions:**

See Weakness.

**Ethical Concerns:**

["NO or VERY MINOR ethics concerns only"]

**Final Justification:**

After the rebuttal and discussion process, I am happy to change my decision to accept this paper.

**Limitations:**

Yes, the proposed method, while promising, reveals certain limitations in the Experiments section.

**Paper Formatting Concerns:**

This paper has no major formatting issues.

**Quality:**

3

**Strengths And Weaknesses:**

Strengths:
1. Balancing privacy and efficiency
2. By using logarithmic functions to dynamically adjust sample weights and preserve low-frequency sample information, an average confusion reduction of 11.42% was achieved on datasets such as Haiku and Sonnets, which is particularly better than the performance of hard deletion methods in data scarcity scenarios.
3. MPC protocol based on semi honest model provides provable security
Weaknesses:
1.The innovation of FedRW is mainly reflected in the first soft deduplication implementation and efficiency optimization in federated scenarios (such as pre-processing acceleration), but its protocol, weight mechanism, and parallel strategy heavily rely on existing technologies (PSI, centralized soft deduplication).
2.This framework provides an efficient and scalable solution for privacy sensitive federated training, but further optimization is needed for semantic redundancy processing and dynamic client adaptation capabilities.

---

> ### Author Rebuttal · Authors · 2025-07-30
>
> We appreciate the reviewer for the valuable and constructive comments. We are encouraged by the positive feedback on **novelty**, **efficiency and effectiveness**, **privacy and security guarantees**, **significance of topic**, and the overall clarity of the paper. We hope the following discussions can address the reviewer's remaining questions.
>
> **1. Rely on Techniques**
>
> We sincerely appreciate your recognition of the innovation and originality of FedRW. We agree that FedRW draws inspiration from the centralized soft deduplication. **However, the same problem remains unresolved in the FL setting, where simply applying the existing techniques introduces new privacy challenges.** Compared with multiple rounds of key agreement and encryption in SOTA work, we leverage the PSI primitive as a foundational component to implement **a lightweight variant, tailored for the defined 2PC functionality**, while achieving **the stronger, third-party-free privacy assumption**. We also present **an innovative parallel orchestration strategy** to overcome the inherent computational overhead of MPC protocols in FL. The proposed innovations achieve up to 28.78× speedup in preprocessing.
>
> Thus, we respectfully argue that the core novelty of our work lies in the innovative integration, adaptation, and optimization of these techniques to address a previously unexplored and challenging problem: enabling efficient and privacy-preserving soft deduplication in FL without relying on a trusted third party.
>
>
> **2. Future Optimization**
>
> We appreciate the reviewer's insightful comments for optimization, and we fully agree that these are vital directions for future research.
>
> Regarding the **semantic matching**, this is indeed a significant issue in large-scale real-world datasets for LLMs. We may leverage the representation learning capability of transformer-based architectures to extract semantic duplication.
>
> Regarding the **dynamic client adaptability**, we acknowledge the importance of dynamic scenarios in which clients may join, leave, or exhibit varying computational capabilities. A preliminary idea is to employ Updatable PSI [1] to address the changes in sample weights caused by client arrivals and departures, and this may trade off between functionality and computational overhead. We plan to explore these valuable aspects in our future research.
>
> **References**
>
> [1] Agarwal, Archita, et al. "Updatable private set intersection from structured encryption." *Cryptology ePrint Archive* (2024).
>
> **We sincerely appreciate your thoughtful and well-justified suggestion, as well as the valuable inspiration you have provided regarding future optimization of our work.**

---

> > ### Comment · Reviewer_xmuh · 2025-08-01
> >
> > Thanks for the discussion and it addressed some of my previous concerns.

---

### Official Review · Reviewer_d6ex · 2025-07-02

**Clarity:** 3
**Significance:** 2
**Originality:** 2
**Rating:** 4
**Confidence:** 3

**Summary:**

In FL settings, there is a fundamental dilemma for deduplication: local fails to detect inter-client duplication. Existing method may remove some domain-specific information and introduce additional overheads. This paper propose a reweighting method to softly deduplicate samples without a trusted third party. This framework reweight based on the frequency for a more nuanced control based on the multi-party computation.

**Questions:**

1. Due to the different data collection fields or geographical locations of different clients, cooperation is meaningful. In this case, it is difficult for two clients to have exactly the same samples. Therefore, when the two repeated samples are not exactly the same, is this method still effective?
2. I also have doubts about the significance of data deduplication in real FL scenarios. Is there such a problem in real datasets? Have different clients collected a large number of similar datasets?
3. The experimental settings mention hyperparameters (e.g., learning rates, epochs), but the paper does not discuss sensitivity analyses or how hyperparameter choices impact performance, which could affect generalizability.
4. This paper also did not conduct experiments with the current mainstream models. Therefore, whether partial repetition will still be harmful when using mainstream models is also an unknown question

**Ethical Concerns:**

["NO or VERY MINOR ethics concerns only"]

**Final Justification:**

I am still concerned about the importance of this issue if the model size gets larger, so I will keep my score.

**Limitations:**

See above

**Quality:**

2

**Strengths And Weaknesses:**

Strength:

1. FedRW introduces a pioneering soft deduplication method for FL, avoiding data deletion and preserving dataset integrity, which is a significant advancement over hard deduplication techniques.
2. By eliminating the need for a trusted third party and using secure 2PC with PSI, FedRW ensures robust privacy guarantees, addressing critical vulnerabilities in decentralized settings.

Weakness:
See questions

---

> ### Author Rebuttal · Authors · 2025-07-30
>
> We appreciate the reviewer for the valuable and constructive comments. We are encouraged by the positive feedback on **novelty**, **privacy and security guarantees**, **significance of topic**, and the overall clarity of the paper. We hope the following discussions can address the reviewer's remaining questions.
>
> **1. Non-Exact Duplicates**
>
> We greatly appreciate this insightful question. Indeed, there are scenarios where non-exact duplicates are more commonly encountered. We have conducted perplexity evaluations on the dataset, *Twitter Sentiment Analysis* [1], which contains non-exact repeated patterns due to the nature of social media.
> - Result
>
>     | Method         | Qwen3-0.6B | Llama-3.2-1B-Instruct |
>     |----------------|-------|-------|
>     | Baseline       |   3.30    |       3.74      |
>     | FedRW (Ours)   |   **2.98**    |    **2.49**         |
>
> These results demonstrate the effectiveness of our method in addressing non-exact repetitive patterns.
>
>
> **2. Significance of Deduplication in Real FL Scenarios**
>
> We truly appreciate this critical concern. Exact and near-exact duplicates are indeed a common issue in large-scale real-world datasets for LLMs. The presence of duplicates among private client data stems from several prevalent mechanisms:
>
> - **Influence of Shared Public Information**: Private data often contains content derived from or heavily influenced by widely shared public information. Similar to open-source datasets like _Twitter Sentiment Analysis_, which inherently exhibits extensive repetition due to the nature of social media, private data collected from users or organizations also tends to contain duplicated or near-duplicate patterns. Specific common patterns may appear across the private data of different clients.
>
> - **Inherent Data Collection**: Beyond external influences, duplicates can occur within sensitive personal information. For instance, GPS locations in personal travel information can exhibit duplicates due to recurring travel patterns or shared high-traffic areas [2], which can be collected to analyze user behavior patterns.
>
> - **Data Augmentation**: The data augmentation strategies [3] used in private datasets, including those based on generative models or synthetic data, may also produce a large number of similar or identical samples with duplicated patterns.
>
> These mechanisms indicate that data duplication is a pervasive challenge in real-world, distributed private datasets in FL, necessitating solutions like FedRW.
>
> **References**
>
> [1] passionate-nlp, "Twitter Sentiment Analysis Dataset." _Kaggle_.
>
> [2] Wu, Zhaomin, et al. "Federated transformer: Multi-party vertical federated learning on practical fuzzily linked data." *Advances in Neural Information Processing Systems 37* (2024): 45791-45818.
>
> [3] Hou, Charlie, et al. "Pre-text: Training language models on private federated data in the age of llms." *arXiv preprint arXiv:2406.02958* (2024).
>
> **3.  Evaluations on Mainstream Models**
>
> **We appreciate your suggestion, and have conducted supplementary perplexity evaluations on Qwen3-0.6B, Qwen2.5-0.5B-Instruct, and Llama-3.2-1B-Instruct under 30% duplication.**
>
> + Qwen3-0.6B
>
>     | Method         | Haiku  | Short Jokes | Rotten Tomatoes | Poetry | Sonnets | Plays |
>     |----------------|--------|-------------|-----------------|--------|---------|-------|
>     | Baseline       |  2.47  |     2.61    |      1.71       |  2.54  |   4.07  |  8.21 |
>     | FedRW (Ours)   |  **2.36**  |     **2.44**    |      **1.59**       |  **2.21**  |   **3.62**  |  **7.23** |
>
> + Qwen2.5-0.5B-Instruct
>
>     | Method         | Haiku | Short Jokes | Rotten Tomatoes | Poetry | Sonnets | Plays |
>     |----------------|-------|-------------|-----------------|--------|---------|-------|
>     | Baseline       |   2.21    |     2.48        |       1.58          |    2.28    |     4.11    |   11.77   |
>     | FedRW (Ours)   |    **2.12**   |      **2.36**       |         **1.55**        |    **2.03**   |   **3.84**    |   **9.92**   |
>
>
> + Llama-3.2-1B-Instruct
>
>     | Method         | Haiku | Short Jokes | Rotten Tomatoes |  Poetry | Sonnets | Plays |
>     |----------------|-------|-------------|-----------------|--------|---------|-------|
>     | Baseline       |   2.14    |     2.34        |     1.65            |         2.39     |    4.11     |  18.35     |
>     | FedRW (Ours)   |    **2.09**   |      **2.21**       |        **1.54**         |          **1.99**    |     **4.00**    |   **16.03**    |
>
>
> As shown in the tables, FedRW consistently maintains its advantage in mitigating the impact of redundancy on model performance, particularly under challenging conditions such as data sparsity or complexity.
>
> **We agree that the impact of partial repetition may vary with model advancements, while data quality remains a fundamental factor in model performance. Deduplication is still an essential preprocessing step to ensure high-quality data.**
>
> **4. Hyperparameter**
>
> **We appreciate this valuable feedback. Regarding epochs, we typically utilize a small number (Appendix C, Line 516) as standard practice to prevent overfitting. Hence, our primary sensitivity analysis focused on the learning rate to assess FedRW's robustness.**
>
> + Setup: We evaluated **perplexity** on DistilGPT2 and Qwen2.5-0.5B-Instruct under **learning rates of 1e-3, 5e-4, 3e-4, 1e-4, 5e-5, and 3e-5**. We selected the **_Plays_** dataset, which exhibited a significant performance gap, to investigate FedRW's generalizability.
>
> + DistilGPT2
>
>     | Method         | 1e-3 | 5e-4 | 3e-4 | 1e-4| 5e-5 | 3e-5 |
>     |----------------|-------|-------|-------|-------|-------|-------|
>     | Baseline       | 16.12 | 16.38 | 16.13 | 14.21 | 15.07 | 14.18 |
>     | FedRW (Ours)   | **14.42** | **14.79** | **14.74** |**13.17** | **14.50** | **12.76** |
>
> + Qwen2.5-0.5B-Instruct
>
>     | Method         | 1e-3 | 5e-4 | 3e-4 | 1e-4| 5e-5 | 3e-5 |
>     |----------------|-------|-------|-------|-------|-------|-------|
>     | Baseline       | 12.86 | 11.07 | 10.63 | 11.50 | 11.77 | 10.67|
>     | FedRW (Ours)   | **11.48** | **9.35** | **8.15** |**8.14** | **9.92**| **9.81** |
>
>
> The experimental results indicate that FedRW consistently maintains its performance advantages and exhibits stable training behavior across this range of learning rates.
>
>
> **We are deeply grateful for your thoughtful consideration, which has enriched our understanding and improved the quality of this work.**

---

### Official Review · Reviewer_L3R8 · 2025-07-03

**Clarity:** 3
**Significance:** 3
**Originality:** 3
**Rating:** 4
**Confidence:** 2

**Summary:**

This paper proposes FedRW, a novel framework for privacy-preserving soft deduplication in federated LLM training. The PPMPR protocol avoids the need for a trusted third party, scales efficiently through a parallel orchestration strategy, and integrates sample-level loss weighting in training.

**Questions:**

Check Weaknesses

**Ethical Concerns:**

["NO or VERY MINOR ethics concerns only"]

**Final Justification:**

Thanks for the clarification. I will remain positive and stick with my original score.

**Limitations:**

Yes

**Paper Formatting Concerns:**

None.

**Quality:**

3

**Strengths And Weaknesses:**

Strengths

The technical contributions are clearly defined, and the paper is well-written with a logical structure.

Weaknesses

1.	It would be helpful if the authors could include training loss curves or convergence comparisons to illustrate whether the proposed reweighting mechanism slows down or accelerates model convergence relative to deletion-based approaches.

2.	There are a few writing-related issues that could improve the clarity of the manuscript. First, using vague terms such as “nuanced” is discouraged in scientific writing, as it does not convey precise meaning. Second, Figures 1 and 2 are visually engaging but lack adequate explanation of key elements. Third, the system formulation in Section 3 is not fully adapted to the federated learning setting, it would benefit from a clearer illustration.

3.	The evaluation is limited to synthetic duplication. It would strengthen the paper to include results or discussion on how FedRW performs in real-world scenarios with naturally occurring duplication, such as instruction-tuning data, which often contains repeated patterns.

4.	The protocol assumes a semi-honest threat model. However, in practice, malicious clients may attempt to manipulate the protocol by misreporting sample frequencies or injecting misleading data into the intersection process. It would be valuable to discuss whether FedRW could be extended or adapted to remain robust under such adversarial behavior.

---

> ### Author Rebuttal · Authors · 2025-07-30
>
> We appreciate the reviewer for the valuable and constructive comments. We are encouraged by the positive feedback on **novelty**, **efficiency and effectiveness**, **privacy and security guarantees**, **significance of topic**, and the overall clarity and logical structure of the paper. We hope the following discussions can address the reviewer's remaining questions.
>
> **1. Convergence Comparisons.**
>
> **We appreciate your suggestion and have included the average training loss of FedRW compared with the baseline under 30% duplication in the tables.**
>
> + GPT-2 Large
>
>     | Dataset         | Method     | Round 0 | Round 1 | Round 2 | Round 3 |
>     |-----------------|------------|---------|---------|---------|---------|
>     | Haiku           | Baseline   |   **1.842** |   0.839 |   0.806 |   0.811 |
>     |                 | FedRW (ours)|   4.671 |   **0.835** |   **0.725** |   **0.705** |
>     | Rotten Tomatoes | Baseline   |   **1.124** |   0.444 |   0.426 |   0.379 |
>     |                 | FedRW (ours)|   3.038 |   **0.425** |   **0.378** |   **0.375** |
>     | Short Jokes     | Baseline   |   **0.994** |   0.720 |   0.622 |   0.641 |
>     |                 | FedRW (ours)|   1.263 |   **0.679** |   **0.595** |   **0.547** |
>     | Sonnets         | Baseline   |   **2.835** |   1.466 |   1.366 |   1.730 |
>     |                 | FedRW (ours)|   4.329 |   **1.349** |   **1.243** |   **1.168** |
>
>
> + DistilGPT2
>
>     | Dataset         | Method       | Round 0 | Round 1 | Round 2 | Round 3 |
>     |-----------------|--------------|---------|---------|---------|---------|
>     | Haiku           | Baseline     |   3.225 |   1.283 |   1.267 |   1.274 |
>     |                 | FedRW (ours) |   **2.950** |   **1.274** |   **1.255** |   **1.252** |
>     | Short Jokes     | Baseline     |   1.511 |   0.751 |     -   |     -   |
>     |                 | FedRW (ours) |   **1.388** |   **0.745** |     -   |     -   |
>     | Rotten Tomatoes | Baseline     |   1.771 |   0.607 |     -   |     -   |
>     |                 | FedRW (ours) |   **1.338** |   **0.437** |     -   |     -   |
>     | IMDB            | Baseline     |   4.234 |   1.964 |     -   |     -   |
>     |                 | FedRW (ours) |   **3.838** |   **1.939** |     -   |     -   |
>     | Poetry          | Baseline     |  10.858 |   2.989 |   1.313 |   1.107 |
>     |                 | FedRW (ours) |   **9.908** |   **1.999** |   **1.122** |   **1.049** |
>     | Sonnets         | Baseline     |  17.589 |   3.443 |   2.019 |   1.870 |
>     |                 | FedRW (ours) |  **15.901** |   **3.048** |   **1.971** |   **1.802** |
>     | Plays           | Baseline     |  23.848 |   3.379 |   2.885 |   2.815 |
>     |                 | FedRW (ours) |  **20.460** |   **3.234** |   **2.854** |   **2.767** |
>
>
> The datasets, including *Short Jokes*, *Rotten Tomatoes*, and *IMDB*, were trained for fewer rounds to mitigate overfitting. Notably, on GPT-2 Large, FedRW exhibited a higher initial loss compared with the baseline, whereas the DistilGPT-2 model showed a lower one. This consistency is related to the loss reweighting mechanism and the variance of model capacity. In summary, FedRW demonstrates **faster model convergence** and **better final performance**.
>
>
> **2. Writing-related Issues.**
>
> **We sincerely appreciate your thorough reading, and we will revise the issues according to your comments in the final version.**
>
> To further clarify Figures 1 and 2, we offer the following supplementary explanation:
>
> + Fig. 1: **(a)** illustrates the inherent challenges of global deduplication in a decentralized setting, where privacy constraints prevent direct data sharing.
> **(b)** describes the paradigm adopted by the state-of-the-art solution (also used as our baseline), which relies on a trusted third party for deduplication over encrypted data, followed by decryption. The right part of Fig.1 indicates that this pattern may lead to the loss of key training samples.
> + Fig. 2: The left panel of Fig.2 illustrates the parallel framework of FedRW, where each small square represents a single 2PC protocol interaction. The middle section depicts our reweighting scheme, which preserves the integrity of the dataset. The colors reflect the different weights assigned to samples. The right panel provides a comparison between FedRW and the baseline approach.
>
> As you pointed out, **Section 3** and **Appendix A.3** illustrate the security and threat model. Due to the space constraints, we were unable to illustrate adaptation to FL in detail. We will include a brief description in the revision as follows:
>
> + FedRW operates under the **semi-honest adversary model** in federated learning (FL), meaning all participants will follow the protocol honestly but may attempt to infer additional information from observation. For the scope of this work, we assume that the parties do not actively collude with one another. While we acknowledge that other forms of more active or malicious attacks exist in FL – such as inference attacks, backdoor attacks, or reconstruction attacks – these threats are considered orthogonal to the primary objective of this study, which focuses on designing the first privacy-preserving soft deduplication via sample reweighting. These additional and vital security considerations will be explored in our future research to enhance FedRW's robustness.
>
> Following your valuable suggestion, we will revise the manuscript with the above information.
>
>
>
> **3. Evaluation in Real-World Scenarios.**
>
> **We appreciate your valuable suggestion, and have conducted experiments on real-world datasets with repeated patterns on two mainstream models, Qwen3-0.6B and Llama-3.2-1B-Instruct.**
> + _databricks-dolly-15k_ [1] is an open-source dataset of instruction-following records generated by thousands of Databricks employees.
> + _Twitter Sentiment Analysis_ [2] is an open-source dataset of tweets labeled with sentiment categories, which inherently contains repeated patterns due to the nature of social media.
>
> - Results (Perplexity)
>
>     - Qwen3-0.6B
>
>         | Method         | databricks-dolly-15k | Twitter Sentiment Analysis |
>         |----------------|-------|-------------|
>         | Baseline       |   5.11    |     3.30        |
>         | FedRW (Ours)   |   **4.65**    |     **2.98**        |
>
>     - Llama-3.2-1B-Instruct
>
>         | Method         | databricks-dolly-15k | Twitter Sentiment Analysis |
>         |----------------|-------|-------------|
>         | Baseline       |   4.79    |      3.74       |
>         | FedRW (Ours)   |   **4.51**    |      **2.49**       |
>
> As presented in the table, FedRW consistently outperforms the baseline under the same experimental settings, showing the effectiveness and generalization ability of our method.
>
> **References**
>
> [1] databricks, "databricks-dolly-15k." *Hugging Face*.
>
> [2] passionate-nlp, "Twitter Sentiment Analysis Dataset." _Kaggle_.
>
> **4. Extension to Malicious Model.**
>
> Thank you for raising the thoughtful concerns about malicious clients. Extending FedRW to resist malicious adversaries would be an interesting and important research direction.
>
> As mentioned in **2**, we acknowledge that the FedRW primarily operates under a semi-honest threat model, which is **sufficient for the vast majority of real-world scenarios**, given its high efficiency and practicality.
>
> Regarding the **malicious model**, it is feasible to design specific verification based on FedRW’s existing architecture, such as **Zero-Knowledge Proofs** (ZKPs), to verify the consistency of parties’ behaviors during preprocessing and training. However, such approaches would come at a significant computational and communication cost.
>
> These potential schemes trade off between **model accuracy**, **data privacy**, and **computational overhead**. Given the scope and depth of these problems, we have deferred this research to our future follow-up work.
>
> **We are deeply grateful for the thought-provoking discussions and valuable insights that have enriched our understanding and improved the quality of this work.**

---

> > ### Comment · Reviewer_L3R8 · 2025-08-07
> >
> > Thank you for the thorough response, it resolves my concerns.

---

### Note · Authors · 2025-08-11

Dear All Reviewers and Area Chair:

We sincerely thank the reviewers for their time and thoughtful engagement, and the conference for its commitment to a fair and constructive review process. We also thank the Area Chair for overseeing the process with diligence and care.

At this stage, all four reviewers have indicated that their concerns were sufficiently addressed, with no unresolved questions remaining. To assist in decision-making, we summarize key points below:

### Strengths Acknowledged by Reviewers
- Reviewers `L3R8`, `d6ex`, and `xmuh` highlighted the **novelty and enhanced privacy guarantees** of our framework.
- Reviewers `L3R8`, `d6ex`, and `y8JQ` recognized the **significance of the addressed challenge**.
- Reviewers `L3R8` and `xmuh` acknowledged the **efficiency and scalability** of our solution.
- All reviewers commended the paper's overall **clarity and logical structure**.

### Major Concerns and Our Response

- Reviewers `L3R8`: Raised four issues, including `convergence comparison`, `real-world duplication`, `malicious model`, and minor `writing issues`. Following our explanations and experiments, Reviewer **confirmed their concerns were resolved**.
- Reviewers `d6ex`: Raised four initial issues, including  `non-exact duplicates`, `prevalence in real FL scenarios`, `mainstream models`, and `hyperparameter sensitivity`. During the discussion, Reviewer raised new concerns about the importance of our method for `larger-scale models`. After our explanations and experiments, Reviewer **confirmed their problems were solved** and **revised the score**.
- Reviewers `xmuh`: Raised two issues, including  `reliance on techniques` and `future optimization`. After the discussion, Reviewer **acknowledged our explanations to address their problems**.
- Reviewers `y8JQ`: Raised three issues, including  `clarity`, `FL settings`, and `mainstream models`. Following our clarifications and experiments, Reviewer **confirmed their concerns were addressed** and **updated the score**.

We have addressed all points in detail in our response to each reviewer. We deeply appreciate all reviewers for their constructive feedback and the AC for diligently overseeing the review process. We will further improve the manuscript based on the insightful suggestions.

Thank you again for your time and consideration.

Best regards,

Authors of Paper #1144

---

### Decision · Program_Chairs · 2025-09-17

**Decision:**

Accept (poster)

**Comment:**

There is a fundamental dilemma for deduplication in FL: local devices fail to detect inter-client duplication. Existing method may remove some domain-specific information and introduce additional overheads. This paper propose a reweighting method to softly deduplicate samples without a trusted third party.

Strenghts: FedRW introduces a new soft deduplication method for FL. By eliminating the need for a trusted third party and using secure 2PC with PSI, FedRW ensures robust privacy guarantees.

Weaknesses: Whether deduplication is a real problem in FL is not fully addressed.

The main reason for my decision is that the reviewers seem to agree on the merits of this paper: proposing a novel solution to deduplication in FL that does not require a trusted 3rd party.